# Launching a saliva-based SARS-CoV-2 surveillance testing program on a university campus

Alexander J. Ehrenberg[1,2], Erica A. Moehle[1,2], Cara E. Brook[1,2], Andrew H. Doudna Cate[1], Lea B. Witkowsky[1,2], Rohan Sachdeva[1,2], Ariana Hirsh[1,2], Kerrie Barry[3], Jennifer R. Hamilton[1,2], Enrique Lin-Shiao[1,2], Shana McDevitt[1,2], Luis Valentin-Alvarado[1,2], Kaitlyn N. Letourneau[1], Lauren Hunter[1], Amanda Keller[2], Kathleen Pestal[1], Phillip A. Frankino[1], Andrew Murley[1], Divya Nandakumar[1,2], Elizabeth C. Stahl[1,2], Connor A. Tsuchida[1,2], Holly K. Gildea[1], Andrew G. Murdock[1,2], Megan L. Hochstrasser[2], Elizabeth O'Brien[1,2], Alison Ciling[1,2], Alexandra Tsitsiklis[1], Kurtresha Worden[1,2], Claire Dugast-Darzacq[1], Stephanie G. Hays[1], Colin C. Barber[1], Riley McGarrigle[1,2], Emily K. Lam[1], David C. Ensminger[1], Lucie Bardet[2], Carolyn Sherry[2], Anna Harte[1,4], Guy Nicolette[1,4], Petros Giannikopoulos[2], Dirk Hockemeyer[1,2,5], Maya Petersen[1], Fyodor D. Urnov[1,2], Bradley R. Ringeisen[1,2], Mike Boots[1], Jennifer A. Doudna[1,2,6]*, on behalf of the IGI SARS-CoV-2 Testing Consortium[¶]

1 University of California, Berkeley, California, United States of America, 2 Innovative Genomics Institute, University of California, Berkeley, California, United States of America, 3 Joint Genome Institute, Lawrence Berkeley National Laboratory, Berkeley, California, United States of America, 4 University Health Services, University of California, Berkeley, California, United States of America, 5 Chan Zuckerberg Biohub, San Francisco, California, United States of America, 6 Howard Hughes Medical Institute, University of California, Berkeley, California, United States of America

¶ Membership of the IGI SARS-CoV-2 Testing Consortium is listed in the Acknowledgments.
* doudna@berkeley.edu

**Data Availability Statement:** All relevant data are within the paper and its Supporting Information files.

## Abstract

Regular surveillance testing of asymptomatic individuals for SARS-CoV-2 has been center to SARS-CoV-2 outbreak prevention on college and university campuses. Here we describe the voluntary saliva testing program instituted at the University of California, Berkeley during an early period of the SARS-CoV-2 pandemic in 2020. The program was administered as a research study ahead of clinical implementation, enabling us to launch surveillance testing while continuing to optimize the assay. Results of both the testing protocol itself and the study participants' experience show how the program succeeded in providing routine, robust testing capable of contributing to outbreak prevention within a campus community and offer strategies for encouraging participation and a sense of civic responsibility.

## Introduction

Routine testing of individuals for the presence of viral genetic material is a central component of pandemic control when the pandemic features asymptomatic or presymptomatic infectious individuals. At the beginning of the SARS-CoV-2 outbreak in the United States, many colleges and universities sought to implement testing procedures for campus communities to detect

**Funding:** We thank the Packard Foundation, the Curci Foundation, the Julia Burke Foundation, and other anonymous donors for their support of IGI FAST. We additionally thank the University of California, Berkeley for their financial support of IGI FAST. A.J.E. is a graduate research fellow at the Greater Good Science Center at the University of California, Berkeley (https://greatergood.berkeley.edu/). J.R.H. is a Fellow of The Jane Coffin Childs Memorial Fund for Medical Research (https://www.jccfund.org/). A.M. is a fellow of the Damon Runyon Cancer Research Foundation (https://www.damonrunyon.org/). The funders had no role in study design, data collection and analysis, decision to publish, or preparation of the manuscript.

**Competing interests:** The authors have read the journal's policy and the authors of this manuscript have the following competing interests: The Regents of the University of California have patents issued and pending for CRISPR technologies on which JAD is an inventor. JAD is a co-founder of Caribou Biosciences, Editas Medicine, Scribe Therapeutics, Intellia Therapeutics, and Mammoth Biosciences. JAD is a scientific advisory board member of Caribou Biosciences, Intellia Therapeutics, eFFECTOR Therapeutics, Scribe Therapeutics, Mammoth Biosciences, Synthego, Algen Biotechnologies, Felix Biosciences, and Inari. JAD is a Director at Johnson & Johnson and Tempus Labs and has research projects sponsored by Biogen, Pfizer, AppleTree Partners, and Roche. FDU is a co-founder of Tune Therapeutics. PG is a co-founder and Director at NewCo Health. PG is the CLIA Laboratory Director for Coral Genomics and 3DMed. This does not alter our adherence to PLOS ONE policies on sharing data and materials.

infectious individuals not presenting COVID-19 symptoms. In March 2020, the Innovative Genomics Institute (IGI) at the University of California, Berkeley launched a high-complexity, Clinical Laboratory Improvement Amendments (CLIA)-certified laboratory to perform clinical SARS-CoV-2 testing for both the campus and surrounding local communities [1]. Initially, the lab developed and validated an automated, mid-turbinate-oropharyngeal swab-based RT-qPCR clinical assay to detect SARS-CoV-2. This laboratory-developed test (LDT) has provided high-throughput clinical testing that supports patients utilizing the campus University Health Services (UHS) and multiple external community healthcare organizations.

In parallel with this effort, the IGI SARS-CoV-2 Testing Consortium designed and implemented an asymptomatic surveillance testing program in summer 2020 to serve the essential facilities and infrastructure staff, and researchers working on the University of California, Berkeley campus, as well as undergraduate students who would return to the campus in the Fall. The program was administered as a research study, enabling us to launch surveillance testing ahead of clinical implementation while continuing to optimize our assay. Here, we provide an account of the steps taken to develop, launch, and optimize the IGI Free Asymptomatic Saliva Testing (FAST) study. We describe our operational successes and limitations, as well as feedback from participants. Together with a companion methodology paper [2] describing the development and validation of the saliva test used for IGI FAST, we provide a roadmap to launching an asymptomatic surveillance program on a university campus.

## Methods

### Enrollment and participation

Recruitment, enrollment, consent, and participation for IGI FAST was approved by the Office for Protection of Human Subjects at the University of California, Berkeley under IRB #2020-05-13336. Informed consent and enrollment were completed on the IGI FAST web application instead of in writing, as a COVID-19 protocol to minimize the need for physical interaction. This web-based consenting step was approved by the IRB. Participants were recruited via email, social media posts, flyers on the University of California, Berkeley campus, word of mouth, campus website postings, and announcements connected to the required campus symptom screening tool. Participants could enroll at any point between June 19, 2020 and October 20, 2020. Enrollment criteria included being at least 18 years of age and affiliation to the University of California, Berkeley campus. Initially, participation was limited to individuals formally approved to work on campus or buildings affiliated with the University of California, Berkeley (e.g., Lawrence Berkeley National Laboratory) as essential workers, including individuals such as visiting scholars, contractors, or regulatory officials who are not formally employees of the University of California, Berkeley but regularly conducted business on campus. This requirement was relaxed in August to allow any individual affiliation to the University of California, Berkeley to enroll, including undergraduate students living off-campus and employees working remotely. Informed consent, flyers, and the study information sheet were available in English and Spanish.

A total of 4,825 participants enrolled in the study; however, 992 did not complete any appointments. A total of 12,602 tests were collected through IGI FAST. From weeks 11–13 of the study (August 31-September 20), there was a pause in study sample collection due to a supply chain shortage of liquid handler pipette tips [2]. Six hundred thirty-one samples were collected during week 11 before the appointment cancellations. Because the majority (n = 586, 93%) of the samples collected during week 11 were affected by the shortage and were unable to be tested, all requisitions from this week are excluded from the analyses we present. These

**Table 1. Demographics of study participants.**

| | Inactive participants[a] | Week 11 participants[b] | Final cohort |
|---|---|---|---|
| n | 992 | 180 | 3,653 |
| Age (mean, SD) | 25.2, 9.9 years | 22.5, 6.8 years | 30.0, 12.2 years |
| Sex (n, %) | Female: 561, 56.6% | Female: 96, 53.3% | Female: 1,964, 53.8% |
| | Male: 422, 42.5% | Male: 83, 46.1% | Male: 1,668, 45.7% |
| | Other: 4, 0.4% | Other: 0, 0.0% | Other: 10, 0.3% |
| | Unspecified: 5, 0.5% | Unspecified: 1, 0.6% | Unspecified: 11, 0.3% |
| Number of tests (n, %) | 0 appointments: 992, 100% | 1 appointment: 180, 100% | 1 appointment: 1,163, 31.8% |
| | | | 2–4 appointments: 1,505, 41.2% |
| | | | 5–8 appointments: 886, 24.3% |
| | | | >8 appointments: 99, 2.7% |

[a]Inactive participants are those who signed up for IGI FAST but, despite not taking any tests, may have taken the exit survey.

[b]Week 11 participants only participated during week 11 when samples were primarily rejected due to supply-chain issues.

exclusion criteria for our analysis leave a final total of 11,971 tests coming from a cohort of 3,653 participants (Table 1) with at least one test in weeks 1–10 and 14–19 of the study.

## Exit survey

All individuals enrolled in the study, except for 13 that withdrew from communications, were requested to take a 15-minute-long exit survey in Qualtrics. The invitation to complete the survey was included in the announcement that IGI FAST would be closing on October 20, 2020 and was available through November 2, 2020. This survey did not solicit identifiable information and was uncoupled from participants' test results. Participants were instructed to skip any questions they did not wish to answer; however, of the total 903 (100%) (Tables 2 and 3) participants that answered at least one question, 847 (94%) completed the whole survey. Because participants could skip questions, and many questions were shown conditionally, we provide sample size on a per-question basis throughout our results here. A text version of the survey is available in S1 Appendix.

## Analysis of testing rates

We sought to compare the results of IGI FAST to expected frequencies based on the estimated prevalence of SARS-CoV-2 infection in the City of Berkeley. To estimate the City of Berkeley's background SARS-CoV-2 prevalence across the duration of IGI FAST, we relied on the 'covidestim' R package developed by Chitwood et al. [3]. We used this package to analyze daily reported COVID-19 cases, deaths, and test positivity rates across the duration of the epidemic for the City of Berkeley, CA (https://data.cityofberkeley.info); the package uses a Bayesian statistical approach and an underlying SIR mechanistic model to infer true infection rates (including asymptomatic) from those reported. The model outputs the estimated true daily infections per 100,000 persons with an upper and lower confidence interval. We then inferred prevalence from these estimated incidence rates as the daily incidence rate multiplied by the duration of infection in days. While an individual infected with SARS-CoV-2 may test positive for more than 21 days [4–6], we used 14 days for this estimate of prevalence, reflective of the pathogen's infectious period [7]. This calculation yielded the estimated true prevalence of infectious cases of SARS-CoV-2 infection per day in our community. We assumed that 40% of this estimated prevalence was composed of asymptomatic or presymptomatic individuals [8, 9].

**Table 2. Characteristics of exit survey respondents (n = 903).**

| | Survey respondents |
|---|---|
| Age (n, %) | 18–24 years: 236, 26% |
| | 25–34 years: 343, 38% |
| | 35–44 years: 118, 13% |
| | 45–54 years: 101, 11% |
| | 55–64 years: 72, 8% |
| | 65+ years: 32, 4% |
| | Unspecified: 1, <1% |
| Gender (n, %) | Woman: 534, 59.1% |
| | Man: 341, 37.8% |
| | Non-binary: 20, 2.2% |
| | Other: 1, 0.1% |
| | Unspecified: 7, 0.8% |
| University role (n, %) | Undergraduate student: 152, 16.8% |
| | Graduate/professional student: 273, 30.2% |
| | Postdoctoral scholar: 95, 10.5% |
| | Non-academic staff: 195, 21.6% |
| | Academic faculty/staff: 186, 20.6% |
| | NA: 2, 0.2% |
| Number of tests (n, %) | 0 appointments: 4, 0.4% |
| | 1 appointment: 84, 9.3% |
| | 2–4 appointments: 337, 37.3% |
| | 5–8 appointments: 346, 38.3% |
| | >8 appointments: 100, 11.1% |
| | NA: 32, 3.5% |

Given that symptomatic individuals were instructed to seek testing at a clinic, we then multiplied the estimated true prevalence (and corresponding confidence intervals) by 40% to yield a daily prevalence of asymptomatic/presymptomatic infections. Finally, we multiplied this asymptomatic/presymptomatic prevalence by the total number of tests collected by IGI FAST each day to determine the expected number of positives per day (S1 Table). To compute the estimated asymptomatic and presymptomatic prevalence and expected number of positives across the study duration, we summed the expected number of true infections per day and divided these infections by a 700,000-person scaling factor to compute mean incidence. We then multiplied this incidence rate by the 14-day duration of infection and the 40%

**Table 3. Race and ethnicity of exit survey respondents (n = 903).**

| Race | Total respondents (n, % of total) | Respondents reporting multiple races (n, % of group) | Hispanic or Latina/o ethnicity (n, % of group) |
|---|---|---|---|
| American Indian or Alaska Native | 7, 0.7% | 7, 100% | 3, 42.9% |
| Asian | 201, 22.3% | 41, 20.4% | 11, 5.5% |
| Black or African American | 21, 2.3% | 11, 52.4% | 2, 9.5% |
| Native Hawaiian or Other Pacific Islander | 3, 0.3% | 0, 0% | 0, 0% |
| White | 634, 70.2% | 56, 8.8% | 42, 6.6% |
| Not reported | 93, 10.3% | – | 45, 48.4% |
| Total | 903, 100% | 59, 6.5% | 95, 10.5% |

asymptomatic/presymptomatic proportion to derive the mean weekly prevalence of asymptomatic/presymptomatic infection. We replicated this approach for the upper and lower confidence interval estimates of infection. To derive mean asymptomatic/presymptomatic prevalence across the study's duration, we applied the same approach but summed estimated cases across the entire study period, divided by 100,000 persons per day multiplied by the total number of days in the study.

## Results and discussion

### Saliva sample choice and study design

We reasoned that voluntary asymptomatic testing would be most effective if the sample collection was simple, tolerable, inexpensive, did not require physical contact with healthcare workers, and could be tested rapidly and robustly. Saliva presented an attractive solution that could meet these criteria. While the exact sensitivity and specificity of saliva-based PCR tests for diagnosis of SARS-CoV-2 infection remain unclear, there was emerging evidence that saliva testing held a comparable performance to nasopharyngeal swabs [10–15] when we began designing the diagnostic test and cognate research study at the end of spring 2020.

We selected saliva as the sampling medium due to the ability to collect specimens amid a shortage of nasopharyngeal swabs with minimal demand for trained personnel and personal protective equipment. We designed and implemented a saliva specimen collection pipeline that minimized exposure risk and maximized ease of use for participants. In parallel, we developed a high-throughput automated qPCR-based laboratory procedure for testing saliva for the presence of SARS-CoV-2 genetic material [2].

A stochastic branching process model [16] guided our asymptomatic surveillance parameters, including testing frequency and turn-around-time (TAT). Since SARS-CoV-2 tests were a limited resource in the San Francisco Bay Area, we sought to identify a surveillance regime that could effectively mitigate asymptomatic spread while retaining an adequate capacity for more vulnerable populations or medically indicated uses. Though there is an obvious association between higher testing frequency and increased outbreak prevention, our model suggested that when viral prevalence is <1% in the participating population, testing participants on alternating weeks with a TAT of no more than five days could limit campus outbreaks. These parameters afforded the IGI Diagnostics Lab the ability to continue allocating sufficient testing resources to highly vulnerable populations in the local community while suppressing asymptomatic transmission within the campus community.

To test the operational feasibility of this model, optimize our assay, and bring surveillance testing to our campus, we established a research study, known as IGI FAST. Of those enrolled, 47% were recruited through a direct email invitation, 42% through word-of-mouth via a friend/coworker, and 35% through invitations included in the clearance messages sent to campus personnel who completed a daily online symptom screener required for on-site work (Fig 1). Interested individuals were directed to a custom-built online web application, where they provided informed consent as well as demographic and contact information to facilitate communication and follow-up should they test positive or inconclusive. Participants receiving a positive or inconclusive (only one of three SARS-CoV-2 genes detected) result through this research study were called by a clinician and directed to take a follow-up swab-based confirmatory clinical test as soon as possible.

### IGI FAST study operating procedure

The IGI FAST study operated for a total of 16 weeks between June 23, 2020 and October 29, 2020 and processed 11,971 tests (S2 Table) from a total cohort of 3,653 active participants

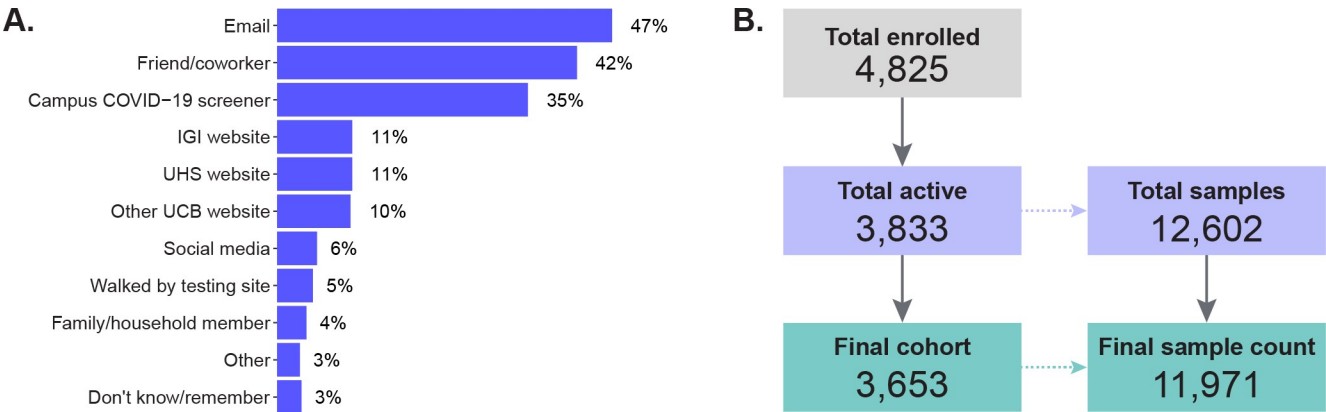

**Fig 1. IGI FAST recruitment.** (A) Results from an exit survey question asking respondents to report through which methods they heard about IGI FAST. Respondents (n = 865) could select multiple answers. Abbreviations: IGI, Innovative Genomics Institute; UHS, University Health Services; UCB, University of California, Berkeley. (B) Of the 4,825 participants who enrolled, 992 did not give any samples, leaving 3,833 participants who gave 12,602 samples. A supply-chain shortage in week 11 caused a large number of samples (94% of samples collected in week 11) to go untested. For this reason, all 631 samples collected at the beginning of week 11 are excluded from the analysis of IGI FAST. One hundred eighty individuals only provided a sample during week 11, so they are excluded from the analysis of IGI FAST, leaving a final cohort of 3,653 participants that gave 11,971 samples in the studied weeks of IGI FAST.

(Fig 1). For the first eight weeks of the study, one outdoor site operated at a campus location near one cluster of research buildings that were the first to resume operations during the pandemic. For the last eight active weeks of the study, an additional site provided expanded access to the on-campus population. Both sites featured the same workflow (Fig 2) which we detail in S1 Methods.

A custom-built IGI FAST web app allowed participants to schedule their saliva collection appointments. Participants received a QR code that was presented and scanned at the testing site, allowing us to rapidly locate participant records in the app. Participants received an SMS text and email reminder 30 minutes before their appointment. Participants were prompted to schedule appointments at a cadence of every two weeks via email. The app uses TLS based encryption, an industry standard for web security, with a Postgres database on top of an AES-256 encrypted filesystem on the backend. The JavaScript files for the enrollment and scheduling app are available at https://github.com/innovativegenomics/igi-testing-kiosk.

In the text and email reminders, participants were instructed not to eat, drink (including water), smoke, chew gum, or brush their teeth for at least 30 minutes before their appointment slot, consistent with instructions from the saliva collection kit manufacturer (DNA Genotek), and were asked to confirm this upon arrival. Participants were screened verbally for COVID-19 symptoms or known exposure. Any individuals reporting symptoms or exposure were instructed to go to UHS, where they were clinically tested using a respiratory swab outside of the IGI FAST study's administration. Those who passed the symptom and exposure screener then scanned their appointment QR code at the check-in desk. Here, they were asked to confirm their name and date of birth. They then received a barcoded saliva collection kit (OMNIgene OM-505).

To provide a saliva test specimen, the participant entered the saliva collection area, where they were directed to an available kiosk staffed by "saliva coach" staff or volunteers who advised on the process from behind a plastic divider. Coaching focused on safety and how to generate an optimal specimen. Kiosk workers observed the saliva sample to check for visible food particles, excessive color (thought to be due to substances such as coffee), excessive mucus, or excessive or insufficient volume (S2 Appendix).

At the intake station (Station 4, Fig 2), participants scanned their tube into a Salesforce-platform laboratory information management system built by Thirdwave Analytics [1] and left the

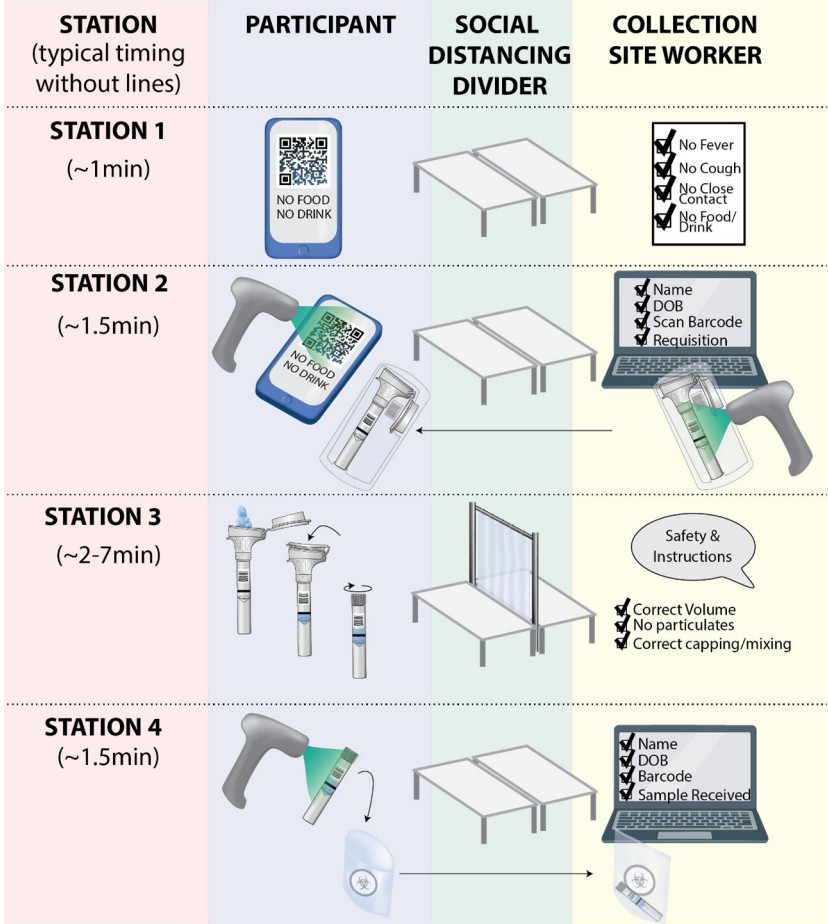

| STATION (typical timing without lines) | PARTICIPANT | SOCIAL DISTANCING DIVIDER | COLLECTION SITE WORKER |
|---|---|---|---|
| **STATION 1** (~1min) | NO FOOD NO DRINK | | ☑ No Fever ☑ No Cough ☑ No Close Contact ☑ No Food/Drink |
| **STATION 2** (~1.5min) | NO FOOD NO DRINK | | ☑ Name ☑ DOB ☑ Scan Barcode ☑ Requisition |
| **STATION 3** (~2-7min) | | | Safety & Instructions ☑ Correct Volume ☑ No particulates ☑ Correct capping/mixing |
| **STATION 4** (~1.5min) | | | ☑ Name ☑ DOB ☑ Barcode ☑ Sample Received |

**Fig 2. IGI FAST testing site workflow.** Our two testing sites followed the same workflow, which can be easily scaled to increase the throughput. Participants first checked in at station one, where they were screened for COVID-19 symptoms, potential contacts, and recent food or water consumption. At station 2, an appointment QR code was scanned, and a collection site worker created a test requisition. Participants then went to individually-staffed spitting stations, where they were supervised while giving their sample. At station 4, the tube was re-scanned, and participants confirmed their name and date of birth. At this point, the sample was dropped into a bag and left with the site worker. Parts of figure made using biorender.com.

tube with the station worker. From here, the samples were brought back to the IGI Diagnostics Lab, where they underwent the procedures outlined in [2].

We produced an instructional video to prepare participants in the test's ongoing clinical deployment (https://youtu.be/2IFB2Q-zV8g). (Note: This video records an IGI FAST test site worker going through the workflow for demonstrative purposes–not for participation in the study).

To protect participants' and workers' safety, we collaborated with University of California, Berkeley's Environmental Health & Safety department and UHS to implement several safeguards. First, we chose to establish the two collection sites outdoors in covered, well-ventilated areas. Ground markings were put in place to encourage social distancing as participants waited in line, and the spitting stations were each separated by >12 feet to prevent the spread of SARS-CoV-2 for the brief period while participants were not wearing masks. Additionally, participants were scheduled to provide specimens at intervals that would keep the density of people low. We established physical barriers between participants and collection site workers

in areas where participants were without masks, and a minimum of six feet of distance elsewhere protected test site workers. Through this attention to safety for both the collection site workers and the participants, we were able to minimize worker demand of personal protective equipment (PPE) while maximizing participant throughput.

On several occasions during the Fall, the Northern California Wildfires created hazardous environmental conditions. If the air quality index, as determined by airnow.gov, was greater than 150 at 8 am on any day, the testing sites were closed, and participants were notified via email and SMS text that no specimens would be collected that day.

Although the IGI FAST test results were not considered protected health information, we nonetheless operated in alignment with HIPAA standards. Results were sent from the berkeley.edu-supported study email address using end-to-end encryption from Virtru. Since these results were from a test yet to be validated under the CLIA framework, we included specific language in the results describing this limitation (S3 Appendix). Participants with positive or inconclusive results were additionally contacted via phone by one of the study clinicians within minutes to several hours following the lab reporting the result to the clinicians. It became critical to quickly recommend isolation to these individuals, follow up with confirmatory testing external to the study, and manage any symptoms that may have emerged.

IGI FAST ended once the assay completed clinical validation as an LDT, obviating the need to administer it as a research study. UHS assumed responsibility for asymptomatic surveillance sample collection to consolidate surveillance sampling resources and personnel on the University of California, Berkeley campus. While we continue to run the saliva test in our laboratory as a clinically orderable test, it is currently deployed in a limited capacity, where its ease of use in a take-home setting better suits low compliance or off-campus student populations. Instead of saliva, a self-administered nasal swab tested on the same PCR-based platform as the IGI FAST test is used for widespread regular asymptomatic surveillance testing at the University of California, Berkeley because these swab-based samples were more easily pooled than the saliva samples.

## Testing and participant characteristics

IGI FAST collected a total of 11,971 tests from its final cohort (S3 Table). We identified five positive samples from five different individuals through IGI FAST, within the expected range of 3.6–33.5 positives predicted by the estimated asymptomatic and presymptomatic prevalence of SARS-CoV-2 infection in the City of Berkeley, CA during IGI FAST's duration (Table 4). While our tested positivity rate falls within the expected range, it is on the lower end. This outcome could be attributable to several factors, including test sensitivity; however, we speculate that it likely reflects a difference in demographics and associated exposure risk between our campus' study cohort and the broader population of the City of Berkeley. Overall, IGI FAST featured a high number (n = 761, 6.4%) of "specimen insufficient" results, making it a difficult test to further scale through pooled testing [2].

Early on, the program confronted insufficient specimen rates up to 39% on a given collection day. In the first three weeks of the program, 376 of the 1,417 (27%) total samples received a specimen insufficient result. While aspects of the lab protocol were optimized to decrease the rate of specimen insufficient results, 158 (42%) of the 376 insufficient specimens were due to protocol failure in a step before PCR, such as an inability to pipette the sample.

Several established challenges with saliva collection may have contributed to this rejection rate. Person-to-person variation in salivary flow, pH, and oral hygiene can contribute to notable heterogeneity in specimen quality [17]. Additionally, several commonly used drugs for hypertension, depression, allergies, pain, inflammation, and recreational use are negatively associated with saliva production, which may lead to repeated sample rejection or difficult

**Table 4. Results and estimated community asymptomatic and presymptomatic prevalence by week.**

| Collection Week[a] | IGI FAST data | | | | | Estimated community prevalence |
|---|---|---|---|---|---|---|
| | Positive | Negative | Inconclusive | Insufficient | Total | Asymptomatic/presymptomatic prevalence[b] (Percent, 95% CI[c]) |
| 6/22/2020–6/28/2020 | 0 | 367 | 1 | 126 | 494 | 0.1% (0.04%, 0.35%) |
| 6/29/2020–7/5/2020 | 0 | 254 | 0 | 115 | 369 | 0.14% (0.06%, 0.45%) |
| 7/6/2020–7/12/2020 | 0 | 417 | 2 | 135 | 554 | 0.13% (0.05%, 0.4%) |
| 7/13/2020–7/19/2020 | 0 | 480 | 0 | 25 | 505 | 0.09% (0.04%, 0.3%) |
| 7/20/2020–7/26/2020 | 0 | 531 | 1 | 42 | 574 | 0.07% (0.03%, 0.22%) |
| 7/27/2020–8/2/2020 | 1 | 530 | 0 | 21 | 552 | 0.07% (0.03%, 0.2%) |
| 8/3/2020–8/9/2020 | 1 | 599 | 2 | 10 | 612 | 0.07% (0.03%, 0.22%) |
| 8/10/2020–8/16/2020 | 0 | 565 | 2 | 42 | 609 | 0.08% (0.03%, 0.24%) |
| 8/17/2020–8/23/2020 | 0 | 588 | 4 | 11 | 603 | 0.08% (0.03%, 0.26%) |
| 8/24/2020–8/30/2020 | 1 | 629 | 3 | 69 | 701 | 0.08% (0.04%, 0.26%) |
| 9/21/2020–9/27/2020 | 2 | 1650 | 1 | 36 | 1688 | 0.03% (0.01%, 0.1%) |
| 9/28/2020–10/4/2020 | 0 | 462 | 1 | 17 | 480 | 0.02% (0.01%, 0.08%) |
| 10/5/2020–10/11/2020 | 0 | 1497 | 1 | 65 | 1563 | 0.02% (0.01%, 0.08%) |
| 10/12/2020–10/18/2020 | 0 | 1068 | 2 | 28 | 1098 | 0.03% (0.01%, 0.13%) |
| 10/19/2020–10/25/2020 | 0 | 1077 | 0 | 4 | 1081 | 0.06% (0.02%, 0.21%) |
| 10/26/2020–11/1/2020 | 0 | 470 | 1 | 15 | 486 | 0.1% (0.04%, 0.35%) |
| **Total** | **5** | **11184** | **21** | **761** | **11971** | **0.08% (0.03%, 0.28%)[d]** |

[a]Weeks 11–13 are excluded here due to the supply chain shortage that shut down testing.

[b]Weekly asymptomatic and presymptomatic prevalence was computed by summing the estimated daily new infections per 100,000 output from the 'covidestim' package in R across the seven days of each week, dividing by a 700,000 person scaling factor to produce the weekly incidence rate, then multiplying by a 14-day duration of infectiousness to derive prevalence (see Methods). Finally, estimates were scaled by 40% to yield the asymptomatic and presymptomatic prevalence per week for the duration of the study period.

[c]Chitwood et al. fixed the lower bound of their 95% confidence intervals at the reported case positive rate (lagged by delay time to presentation of symptoms). As a result, confidence intervals are not always evenly distributed (upper bounds exceed lower bounds).

[d]The asymptomatic and presymptomatic prevalence for the entire study duration (6/23/2020–8/30/2020 and 9/21/2020–10/29/2020) was computed similarly as the weekly asymptomatic and presymptomatic prevalence, but summed the estimated daily new infections across the entire study duration, then followed the same steps.

sampling for select individuals [18]. However, we suspected that factors like postnasal drip, improper sample volumes, or contaminants from food or drink significantly contributed to our high sample rejection rate. Accordingly, we established a communication line between the lab and collection site workers to improve coaching and establish more careful on-site sample screening. Together with optimization of the assay in the lab, the more stringent quality control steps occurring at the collection site decreased the specimen insufficient rate from 27% in the first three weeks to 1.8% in the last three weeks (Fig 3).

These changes also improved the TAT for results. Throughout the study, the mean (standard deviation) TAT decreased from 72.6 (68.8) hours in the first three weeks (n = 1,417) of the study to 45.9 (19.6) hours in the last three weeks (n = 2,665) of the study (Fig 3).

The characteristics of our study population are described in Table 1. Overall, 4,825 participants enrolled, with 992 individuals who never gave a specimen and 180 who completed an appointment only during week 11's supply chain shortage, which are not included in the final cohort of 3,653 (Fig 1).

## IGI FAST participant experience assessed by survey

All individuals enrolled in IGI FAST by October 20, 2020 were invited via email to take a 15-minute anonymous exit survey in Qualtrics (S1 Appendix), including those who never

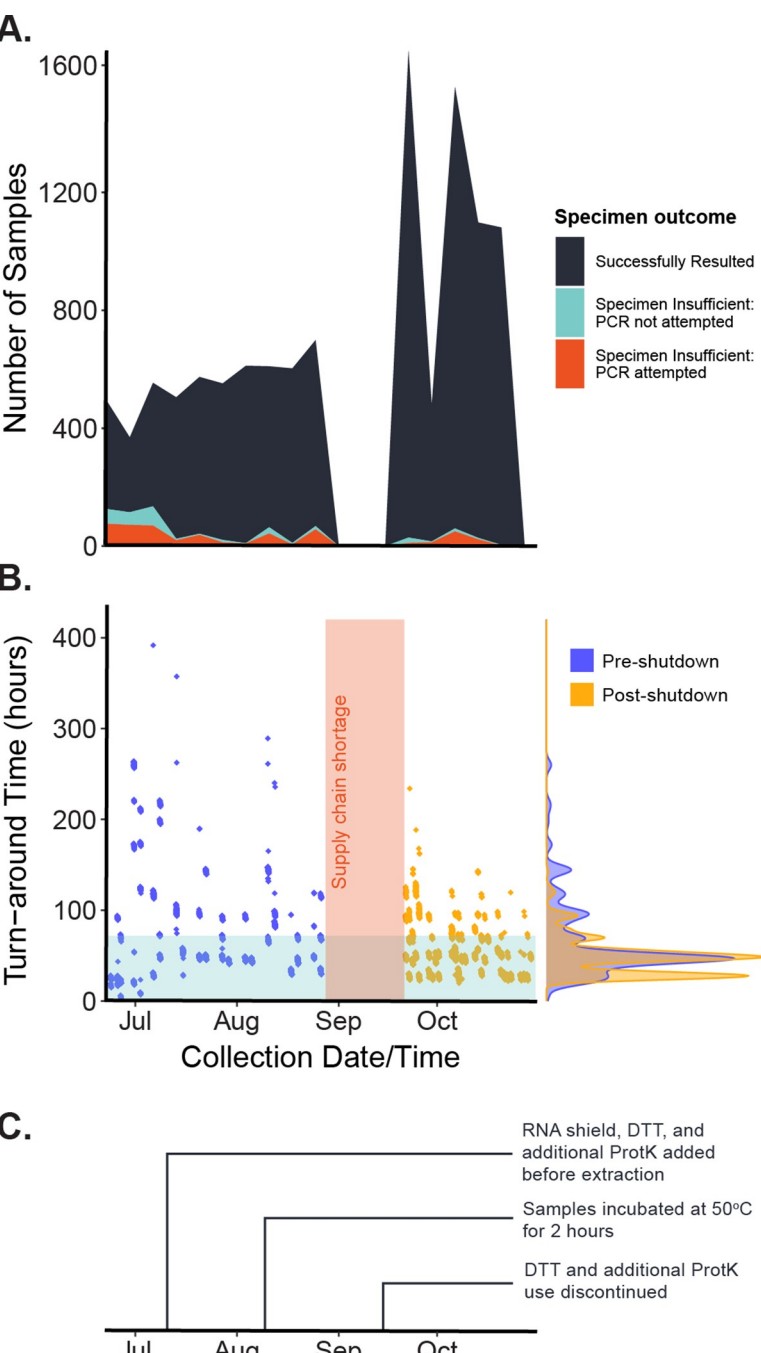

**Fig 3. Sample characteristics throughout the study duration.** (A) The proportion of samples (n = 11,971) that returned with a specimen insufficient value decreased throughout the study. The teal-colored band represents samples that were rejected in a step before PCR analysis, such as an inability to pipette. (B) The mean (standard deviation) turn-around time decreased from 72.6 (68.8) hours in the first three weeks (n = 1,417) of the study to 45.9 (19.6) hours in the last three weeks (n = 2,665) of the study. Here, the blue bar depicts a 72-hour turn-around. The vertical red bar depicts the supply-chain shortage period that led to a temporary shutdown in testing. (C) A series of laboratory techniques (see Hamilton et al.) were deployed throughout the study to optimize the protocols, improving the sample rejection rate and turn-around time.

made an appointment. A total of 903 (19%) (Table 2) participants completed at least one question on the survey (S4 Table). Race and ethnicity data (Table 3) were collected on the exit survey, but not during original testing enrollment. This survey collected data on demography, housing characteristics, behaviors associated with SARS-CoV-2 exposure risk, experiences with IGI FAST, and preferences related to surveillance testing. It also solicited general impressions in the form of the question, "If you had the opportunity to give advice to another university setting up SARS-CoV-2 surveillance testing, what are some suggestions you would give?" Selected excerpts from responses to this question are provided in the next section.

## Key results from the IGI FAST participant survey

### 1. Convenience is critical

*"I could walk out of my building, do my test, and be back in [the] lab in a matter of 10 minutes."–Graduate/professional student*

As a voluntary testing program, we focused heavily on making IGI FAST easy to participate in and widely appealing. Ease of enrollment, quick appointment scheduling, and testing sites' location were paramount. Appointment scheduling could be completed in well under one minute on the study's web application. The workflow (Fig 2) was kept as streamlined as possible on-site. While saliva collection duration was variable, we endeavored to supply sufficient kiosks and coaches to keep the site's total time to under ten minutes. Our locations were chosen, in part, based on their proximity to buildings with high concentrations of on-site personnel– 82% of survey respondents that were approved to work on campus worked within ten minutes of an IGI FAST testing site. Survey responses emphasized the importance of this choice and minimizing travel time, where 58% of exit survey respondents indicated that they would no longer participate if they had to travel longer than ten minutes from their workspace to get to a testing site (Fig 4).

Specimen collection procedures in IGI FAST were well-tolerated and viewed as easy by participants (Fig 5). When asked to compare experiences with SARS-CoV-2 tests received elsewhere, 79% (409 out of 515) of those that received a respiratory swab reported that the IGI FAST saliva test was more tolerable (Fig 5). While there is a strong preference for saliva over clinician-administered respiratory swabs, our exit survey indicates that a switch to self-administered nasal swabs would not significantly affect participation rates. As one graduate student participant commented in the exit survey, ". . .the self-administered nasal swabs are a pain and make me not want to go as much, though [I will] probably put up with [them anyway]." Only 2% of survey respondents indicated that they would not participate in a surveillance program using a self-administered nasal swab, while 86% indicated that they would participate in such a program, and an additional 12% were unsure (Fig 5). Additionally, IGI FAST received high scores for ease and safety (Fig 5). Taking our data together, convenience is the most critical determinant of participation, suggesting that, if made as convenient as FAST, a voluntary nasal swab-based asymptomatic surveillance program is likely to see high participation rates.

### 2. Saliva presents challenges but should not be ignored as an option

*"I really like[d] that IGI FAST used saliva collection instead of a nasal swab and found it much more comfortable than the [self-administered nasal swab] test."–Undergraduate* student

Throughout the FAST study, we identified several challenges with saliva-based testing for SARS-CoV-2. First, it requires advanced planning by the participants to ensure that they do not eat, drink, smoke, chew gum, or brush their teeth within 30 minutes before their

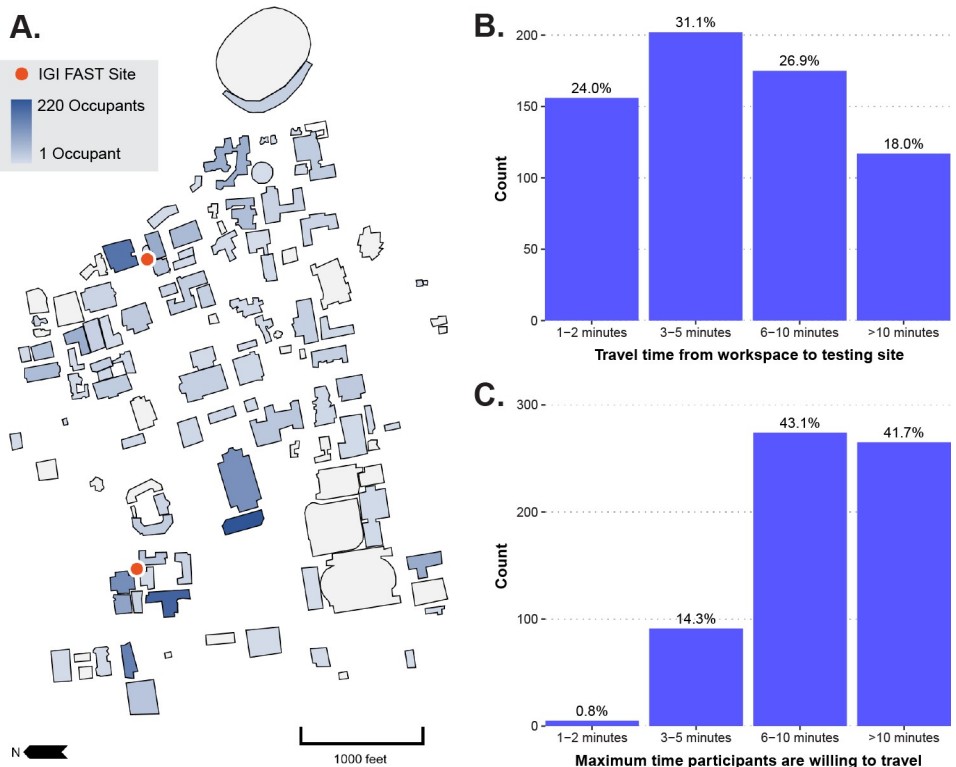

**Fig 4. Convenience of IGI FAST testing sites.** IGI FAST aimed to select testing sites on the University of California, Berkeley campus (A) near the buildings with the highest occupancy levels during the pandemic. Here, we represent occupancy based on answers given during enrollment. Residence halls and off-campus buildings are not depicted; neither are participants who did not report any building. Participants who provided a primary campus building were presumed to be approved to work in those buildings during the pandemic in this illustration. Participants who reported multiple campus buildings are counted multiple times in this illustration. (B) Overall, 82% of survey respondents who were approved to work on campus indicated that they worked within 10 minutes of the nearest testing site, excluding those who did not answer this question on the survey (n = 650). (C) When asked how long they would be willing to travel for regular surveillance testing, most survey respondents reported that they would be willing to travel six or more minutes for testing; however, many participants would be lost to travel times greater than ten minutes (n = 635).

appointment times. While there is limited research on the subject [19], this guidance is typical of saliva kit manufacturers' instructions to prevent interference with the abundance of buccal cell DNA or populations of oral viruses and microbes. Second, we suspect that either natural variability [17], drug- or disease-induced xerostomia [18], or participant hydration status affected the viscosity of samples collected, leading to high variability in sample quality. We found saliva to be a challenging matrix for nucleic acid extraction in general and observed that viscous samples were often more likely to fail, leading to a high specimen insufficient rate. These technical issues are further discussed in our companion manuscript [2], and made the possibility of pooling to increase surveillance capacity impractical, a strategy we were able to test by virtue of operating under an IRB rather than as a clinical requisition. Third, 20% of survey respondents indicated that producing a sufficient amount of saliva was either "somewhat difficult" or "extremely difficult." However, 33% of respondents with two or more visits indicated that they developed a saliva sampling strategy, indicating a possible learning curve. The most common strategies included building a "reserve" of saliva while waiting in line by not swallowing (41% of those with strategies), hydrating well before the test (36%), and thinking about food during the test (12%).

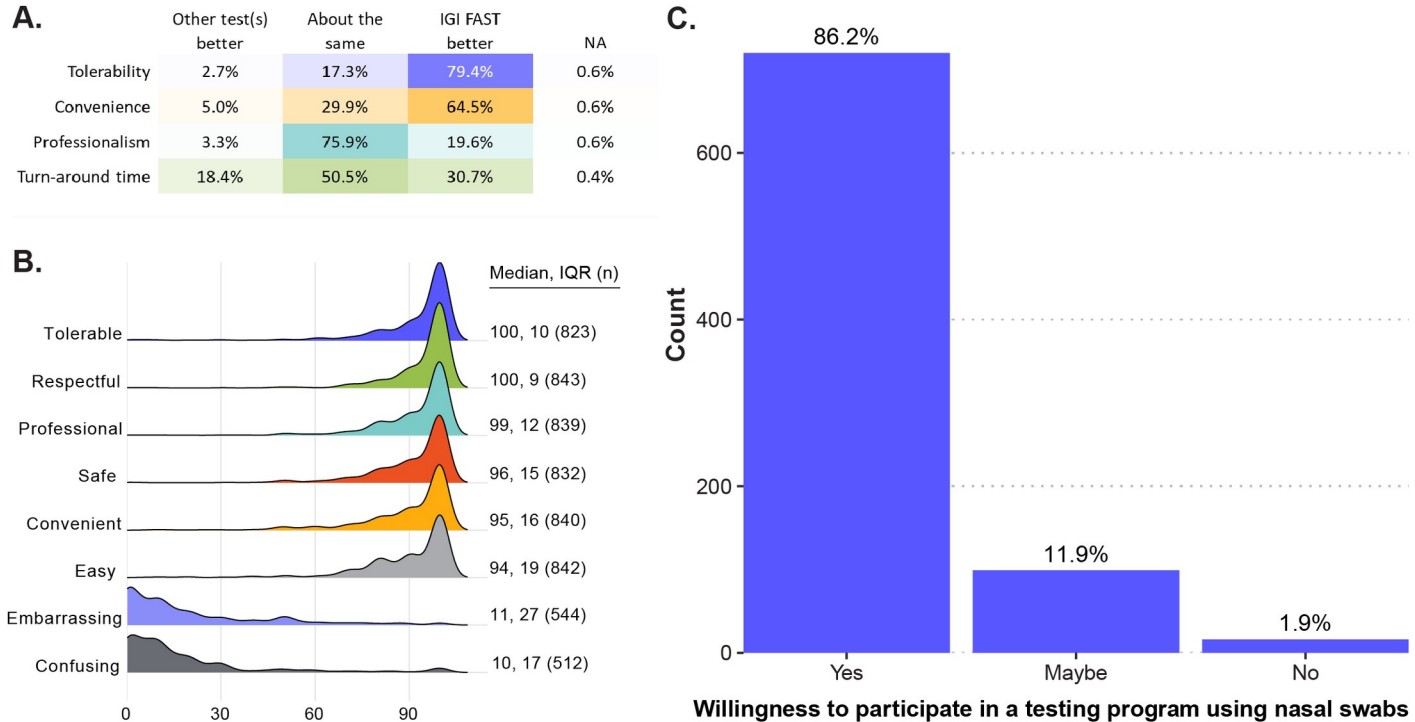

**Fig 5. Participant reviews of IGI FAST.** IGI FAST was well-reviewed by participants. (A) To exit survey respondents who reported having done a respiratory swab-based SARS-CoV-2 test outside of IGI FAST (n = 515), IGI FAST was superior regarding tolerability and convenience. (B) When exit survey participants were asked how well various words described their experiences with the IGI FAST test, testing sites, or personnel, IGI FAST received favorable responses. (C) When asked whether participants would continue participating in a testing program using a nasal swab instead of IGI FAST, most respondents indicated a willingness to continue participation (n = 839). Of note, this question specified the hypothetical continued program would use nasal swabs instead of nasopharyngeal swabs, clarifying the difference between the two.

Despite these challenges, saliva as a sample type retains certain advantages over respiratory swabs. As spitting is a non-technical procedure, saliva samples are particularly amenable to at-home self-collection. Collecting saliva also presents an alternative for populations who are particularly intolerant to respiratory swabs and circumvents shortages in respiratory swab supply chains. To further explore its potential to better reach low-participation populations, the IGI has partnered with UHS to continue a smaller-scale take-home pilot using the now clinically-validated saliva-based assay developed during IGI FAST. To go along with this pilot, IGI has created resources such as video (https://youtu.be/FRuAcLJm5zk) instructional materials for use at home. Given that saliva tests appear to have comparable performance to nasopharyngeal swab tests for SARS-CoV-2 [20, 21], expanding deployment beyond asymptomatic surveillance into some clinical populations may be warranted. In fact, there is emerging evidence in pre-print literature that SARS-CoV-2 titer in saliva may be a helpful biomarker for risk stratification and prognosis [22].

**3. Establish regular testing as a social norm**

*"Create a climate where an employee sees it as something to do for their coworkers[,] not as a threat to their continued employment. . ."–Non-academic research staff member*

Given that participation was entirely voluntary and no compensation was given, our successful enrollment of 3,653 active participants indicates a demand for, rather than resistance to surveillance testing in general. Indeed, a study at the University of California, Berkeley

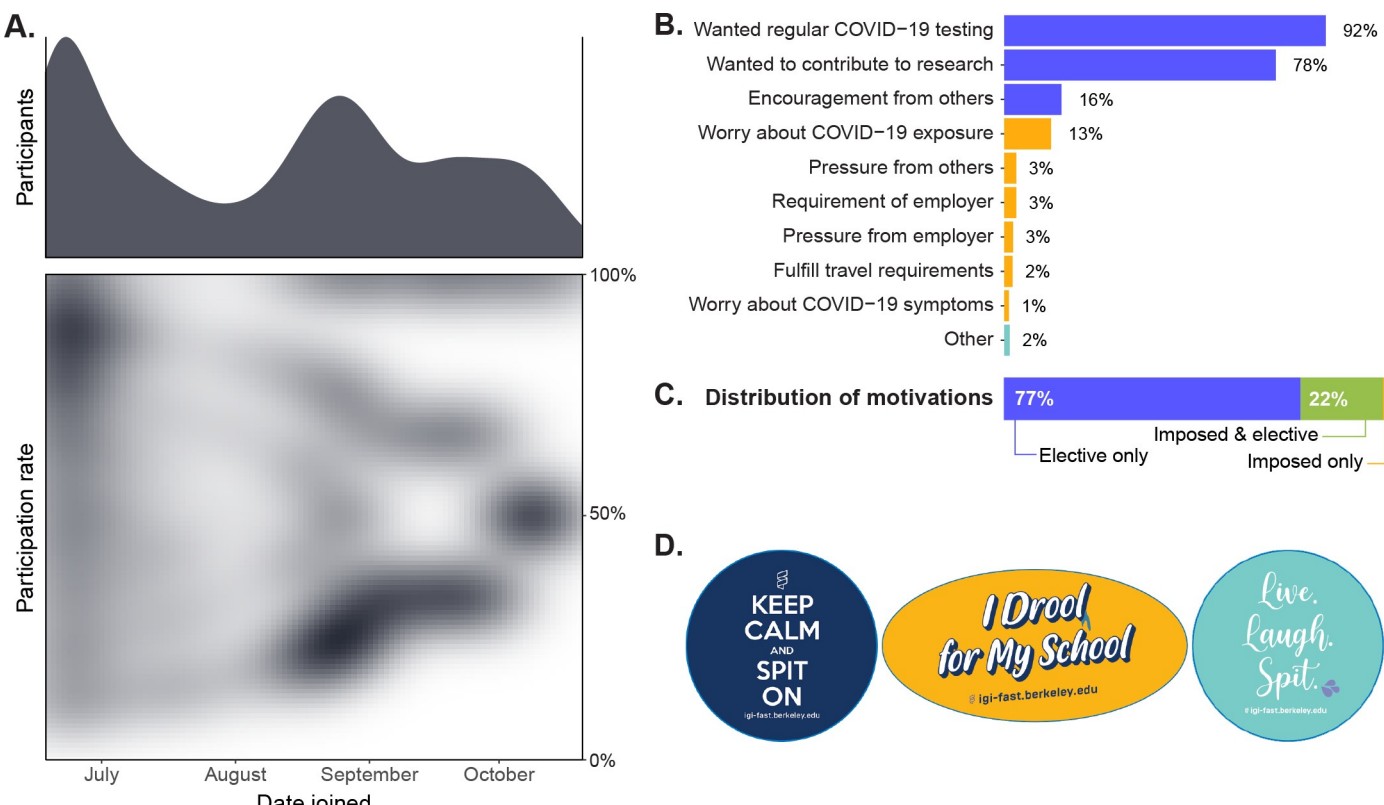

**Fig 6. IGI FAST enrollment.** (A) There were two waves of enrollment into the study–at the beginning of the study and again in the Fall when undergraduate students returned to the city for the beginning of the academic year. There was a wide variety of participation levels in the program. Here, participation rate represents the number of samples an individual gave divided by the number of possible appointments that individual could have made based on when they joined. The graphs here depict only active participants in the final cohort (n = 3,653). (B) The leading reasons for joining IGI FAST were elective (i.e., participants wanted regular viral testing, wanted to contribute to research, or had encouragement from friends/family/coworkers). 79% of the 865 respondents who completed this question reported more than one reason. (C) The majority (77%) of the total (n = 865) participants reported solely elective reasons for joining IGI FAST. Few (1%) reported imposed or "one-off" reasons (i.e., worry about COVID-19 exposure or symptoms, pressure from others, requirement or pressure from employer/boss/ supervisor, the fulfillment of travel requirements). One individual was excluded from this analysis because their response was unclassifiable. (D) Examples of stickers produced to facilitate the normalization of surveillance testing on campus through cultivating a sense of pride.

immediately preceding IGI FAST [23] supports our findings that creating attitudes of civic engagement and camaraderie surrounding surveillance testing may render mandates unnecessary. 648 (17.7%) of the 3,653 active participants participated fully (i.e., took every appointment opportunity offered to them) (Fig 6). The reasons underlying the lack of full participation by 82.3% of participants were not systematically assessed; however, anecdotal accounts by participants cited periods of not conducting on campus work, travel, or self-direction towards clinical testing due to symptoms or exposure as some reasons for missing appointments. Overall, the reasons for joining IGI FAST indicated in the exit survey were typically elective rather than imposed (Fig 6). Of the respondents who were approved to work on the University of California, Berkeley campus at any time during IGI FAST, 90% reported that they either preferred or did not feel safe working unless they and their coworkers got regular viral testing.

We endeavored to cultivate a positive culture surrounding surveillance testing through messaging strategies and testing site atmosphere. Training efforts with IGI FAST personnel focused on creating a welcoming, respectful environment for participants (Fig 5). A specific effort to promote positive messaging surrounding surveillance testing featured a series of

stickers produced by IGI (Fig 6). While subtle, the stickers provided a mode of civic signaling akin to the "I voted" stickers that increase voter turnout in elections by evoking conformity bias [24]. One undergraduate survey respondent reported that the ". . .stickers started a movement of sorts within my house and I felt a sense of pride getting tested." Some participants even reported an effort to collect all sticker designs across their visits. Acknowledging that our survey lacks assessment of individuals not choosing to participate in surveillance testing, we conclude that with appropriate messaging and atmosphere, surveillance testing for SARS--CoV-2 can quickly become a social norm and even be perceived as a civic duty.

**4. Establish a robust communication system**

*"Work with a [communications professional to] write, edit and design your communication materials. We are [drowning] in information, which is changing daily and it does not help if we have to wade through [. . .] communication."–Healthcare worker*

IGI FAST aimed to communicate regularly and clearly with study participants. By creating a website solely dedicated to testing through IGI FAST, participants had an easily accessible, coherent resource to understand how to participate in the program, access appointments, and view announcements with program updates. Collaboration with UHS enabled individuals who had provided positive or inconclusive samples to promptly receive clinical testing and care. Furthermore, an automated notification system enabled participants to receive prompts to schedule appointments and reminders about upcoming appointments, including reminders to avoid eating or drinking before their appointment via email and SMS text.

Future improvements to the testing program could include faster dissemination of results, both to participants and the campus health service, which could be achieved by enhancing the study web app to include a results portal. Other enhancements to the web app could include centralizing resources for contact tracing, symptom screening, and clear, digestible guidance on best practices for social behavior, mask-wearing, and use of asymptomatic testing results. Universities should aim to centralize all resources related to pandemic response and testing and be transparent regarding policies and procedures to minimize stigmatization surrounding positive results.

**5. Risk-stratification is helpful but should be comprehensive**

In a limited-resource setting, campuses benefit from developing a risk-stratification paradigm to determine cost-effective test allocation models. Such an allocation model was not deployed in IGI FAST, as the program was initially designed solely for a population of on-campus workers. Many campus surveillance efforts have focused on undergraduate students living in administratively defined congregate housing (i.e., dormitory residents, those living in Greek or co-operative housing) and actively training or competing student-athletes. Exposure and transmission risk-stratification can be further optimized by considering additional factors, including 'off-campus' housing density and outside work activities. For example, 14% of undergraduate survey respondents (n = 115) live in private households or apartments with greater than five other people, and 5% of graduate student survey respondents who are approved to work on campus live with someone who works or volunteers in healthcare settings or nursing homes. Notably, some participants were concerned about self-isolating should they test positive. Only 50% of graduate/professional students (n = 250) reported that they could "probably" or "definitely" effectively self-isolate in response to a positive test result. A brief questionnaire to collect baseline data such as housing characteristics, risks associated with housemate occupation, childcare/school attendance, ability to self-isolate, or housemate susceptibility to adverse health outcomes could be deployed to allocate resources to the campus population more effectively. Continued brief questionnaires may further refine the model of

resource allocation to assess factors such as social behavior, travel, changes in occupational risk, or even emergence of symptoms [23]. Doing so, with guidance from data-driven epidemiological models such as Brook et al [16], would layer adaptive risk-stratification onto a baseline risk-stratification paradigm to ensure optimal resource allocation longitudinally.

## Limitations

While we aimed to establish a robust testing program and research study, it had several limitations. To facilitate the approval and establishment of the protocol, we did not solicit any health records from participants. This includes the results of any confirmatory testing or data regarding the subsequent emergence of symptoms. While this did not hinder our goal of identifying and directing asymptomatic individuals infected with SARS-CoV-2 to clinical services, this choice limited the breadth of analyses we could conduct here. Furthermore, our exit survey attempts to study the factors influencing participation in SARS-CoV-2 surveillance testing. Seeing that the exit survey was only sent to participants, we do not capture the attitudes of individuals choosing not to participate in surveillance testing in the survey responses. As such, conclusions drawn from the survey responses should be interpreted with caution.

## Conclusion

During the SARS-CoV-2 pandemic, higher education institutions have been faced with difficult choices in their effort to retain some activity while safeguarding their campus and local communities. While many challenges are universal, Universities present some unique risks of spread both within their campus and outside to local communities stemming from student travel between campus and their home locales, engagement in high-contact student athletics, high-density living spaces [25, 26], and a culture of highly social behavior. For these reasons, universities that choose to maintain some in-person activities have an obligation to minimize the spread of SARS-CoV-2 through a robust disease prevention ecosystem. Here we describe a blueprint for a low-barrier, safe, effective, easy, and adaptable program for campus SARS-CoV-2 surveillance using saliva specimens capable of minimizing the number of outbreaks [16] and effectively creating a culture of safety.

Universities have an opportunity to lead with institutional responses to crises. Campus responses to SARS-CoV-2 have had the opportunity to serve as a paragon of effective utilization of academics with expertise in the life sciences, public health, technology, public policy, social psychology, education, and communications. We hope that the outcomes and lessons from IGI FAST will help other institutions implement successful strategies or improve their responses in the face of SARS-CoV-2 and assist with future pandemic preparedness.

## Supporting information

**S1 Appendix. IGI FAST exit survey.**
(PDF)

**S2 Appendix. Visual inspection guide for on-site saliva sample screening.** See https://innovativegenomics.org/wp-content/uploads/2021/01/visual-inspection-guide.pdf for the full-quality version.
(DOCX)

**S3 Appendix. Generic versions of the results emails used for IGI FAST.**
(DOCX)

**S4 Appendix. IGI SARS-CoV-2 Testing Consortium membership.**
(DOCX)

**S1 Methods. Detailed materials and methods for saliva collection sites.**
(DOCX)

**S1 Table. Daily IGI FAST results and estimates of asymptomatic and presymptomatic SARS-CoV-2 infection.** Estimates derived from the 'covidestim' R package calculated from prevalence in the City of Berkeley.
(DOCX)

**S2 Table. IGI FAST samples and associated results.**
(CSV)

**S3 Table. IGI FAST participants.**
(CSV)

**S4 Table. IGI FAST survey responses.**
(CSV)

## Acknowledgments

We thank all the IGI FAST participants for their involvement in the study. Additionally, we are grateful to Robert Tijan (UCB/HHMI); Chancellor Carol Christ, Executive Vice Chancellor Paul Alivisatos, Vice Chancellor for Administration Marc Fisher, Assistant Vice Chancellor for Human Resources Eugene Whitlock, and Vice Chancellor for Research Randy Katz (UCB); Melody Heller and Lisa Polley (UCB University Health Services); Chips Hoai (UCB Environmental Health Services); Chief Margot Bennett (UCB Police) and UCPD Community Service Officers; Emily Harden-Antonio and Adrienne Tanner (UCB Office for Protection of Human Subjects); and Patricia Zialcita (City of Berkeley Public Health) for their contributions, support, and guidance. We extend our gratitude to Benton Cheung from IGI for videography assistance. We thank all the testing staff of the IGI FAST surveillance effort for their support of the initiative.

Membership in the IGI SARS-CoV-2 Testing Consortium is provided in S4 Appendix. The IGI SARS-CoV-2 Testing Consortium is led by Jennifer A. Doudna (doudna@berkeley.edu).

## Author Contributions

**Conceptualization:** Alexander J. Ehrenberg, Erica A. Moehle, Cara E. Brook, Andrew H. Doudna Cate, Lea B. Witkowsky, Rohan Sachdeva, Ariana Hirsh, Kerrie Barry, Jennifer R. Hamilton, Enrique Lin-Shiao, Shana McDevitt, Luis Valentin-Alvarado, Lauren Hunter, Kathleen Pestal, Phillip A. Frankino, Andrew Murley, Divya Nandakumar, Elizabeth C. Stahl, Connor A. Tsuchida, Andrew G. Murdock, Megan L. Hochstrasser, Elizabeth O'Brien, Alison Ciling, Alexandra Tsitsiklis, Kurtresha Worden, Claire Dugast-Darzacq, Stephanie G. Hays, Lucie Bardet, Anna Harte, Guy Nicolette, Petros Giannikopoulos, Dirk Hockemeyer, Maya Petersen, Fyodor D. Urnov, Mike Boots, Jennifer A. Doudna.

**Data curation:** Alexander J. Ehrenberg, Andrew H. Doudna Cate, Rohan Sachdeva, Kaitlyn N. Letourneau.

**Formal analysis:** Alexander J. Ehrenberg, Erica A. Moehle, Cara E. Brook, Lea B. Witkowsky, Kaitlyn N. Letourneau, Mike Boots.

**Funding acquisition:** Lucie Bardet, Jennifer A. Doudna.

**Investigation:** Alexander J. Ehrenberg, Erica A. Moehle, Lea B. Witkowsky, Jennifer R. Hamilton, Enrique Lin-Shiao, Shana McDevitt, Luis Valentin-Alvarado, Amanda Keller, Kathleen Pestal, Phillip A. Frankino, Andrew Murley, Elizabeth C. Stahl, Connor A. Tsuchida, Holly K. Gildea, Elizabeth O'Brien, Alison Ciling, Alexandra Tsitsiklis, Kurtresha Worden, Claire Dugast-Darzacq, Colin C. Barber, Riley McGarrigle, Emily K. Lam, David C. Ensminger, Carolyn Sherry, Anna Harte, Guy Nicolette, Petros Giannikopoulos, Fyodor D. Urnov, Jennifer A. Doudna.

**Methodology:** Alexander J. Ehrenberg, Erica A. Moehle, Cara E. Brook, Andrew H. Doudna Cate, Lea B. Witkowsky, Rohan Sachdeva, Ariana Hirsh, Jennifer R. Hamilton, Enrique Lin-Shiao, Shana McDevitt, Kaitlyn N. Letourneau, Lauren Hunter, Amanda Keller, Kathleen Pestal, Alexandra Tsitsiklis, Kurtresha Worden, Claire Dugast-Darzacq, Stephanie G. Hays, Colin C. Barber, Riley McGarrigle, Emily K. Lam, David C. Ensminger, Petros Giannikopoulos, Dirk Hockemeyer, Bradley R. Ringeisen, Jennifer A. Doudna.

**Project administration:** Alexander J. Ehrenberg, Andrew H. Doudna Cate, Divya Nandakumar, Lucie Bardet, Carolyn Sherry, Petros Giannikopoulos, Bradley R. Ringeisen, Jennifer A. Doudna.

**Resources:** Andrew H. Doudna Cate, Megan L. Hochstrasser, Carolyn Sherry, Petros Giannikopoulos, Fyodor D. Urnov, Bradley R. Ringeisen, Jennifer A. Doudna.

**Software:** Andrew H. Doudna Cate, Rohan Sachdeva.

**Supervision:** Maya Petersen, Fyodor D. Urnov, Bradley R. Ringeisen, Mike Boots, Jennifer A. Doudna.

**Validation:** Cara E. Brook.

**Visualization:** Alexander J. Ehrenberg, Lea B. Witkowsky.

**Writing – original draft:** Alexander J. Ehrenberg, Lea B. Witkowsky, Jennifer A. Doudna.

**Writing – review & editing:** Alexander J. Ehrenberg, Erica A. Moehle, Cara E. Brook, Andrew H. Doudna Cate, Lea B. Witkowsky, Rohan Sachdeva, Ariana Hirsh, Kerrie Barry, Jennifer R. Hamilton, Enrique Lin-Shiao, Shana McDevitt, Luis Valentin-Alvarado, Kaitlyn N. Letourneau, Lauren Hunter, Amanda Keller, Kathleen Pestal, Phillip A. Frankino, Andrew Murley, Divya Nandakumar, Elizabeth C. Stahl, Connor A. Tsuchida, Holly K. Gildea, Andrew G. Murdock, Megan L. Hochstrasser, Elizabeth O'Brien, Alison Ciling, Alexandra Tsitsiklis, Kurtresha Worden, Claire Dugast-Darzacq, Stephanie G. Hays, Colin C. Barber, Riley McGarrigle, Emily K. Lam, David C. Ensminger, Lucie Bardet, Carolyn Sherry, Anna Harte, Guy Nicolette, Petros Giannikopoulos, Dirk Hockemeyer, Maya Petersen, Fyodor D. Urnov, Bradley R. Ringeisen, Mike Boots, Jennifer A. Doudna.

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
