## [Decision Letter · Decision Letter 0]

29 Mar 2021

PONE-D-21-05507

Launching a saliva-based SARS-CoV-2 surveillance testing program on a university campus

PLOS ONE

Dear Dr. Doudna,

Thank you for submitting your manuscript to PLOS ONE. After careful consideration, we feel that it has merit but does not fully meet PLOS ONE’s publication criteria as it currently stands. Therefore, we invite you to submit a revised version of the manuscript that addresses the points raised during the review process.

Authors should consider the comments of both reviewers, especially regarding protocols (Reviewer #1), references, and others (Reviewer #2).

We look forward to receiving your revised manuscript.

Kind regards,

Ruslan Kalendar, PhD

Academic Editor

PLOS ONE

Journal Requirements:

2. Please ensure you have discussed any potential limitations of your study in the Discussion.

3. In the Methods, please clarify that participants provided digital consent. Please also state in the Methods:

- Why written consent could not be obtained

- Whether the Institutional Review Board (IRB) approved use of digital consent

If additional written consent was collected as part of the test sample collection, please include this information.

For more information, please see our guidelines for human subjects research: https://journals.plos.org/plosone/s/submission-guidelines#loc-human-subjects-research

4. Thank you for including your ethics statement: 

"Office of the Protection of Human Subjects, University of California, Berkeley

IRB #2020-05-13336

Written (online) consent was obtained from all participants".   

a. Please amend your current ethics statement to confirm that your named institutional review board or ethics committee specifically approved this study.

5. Thank you for stating in your financial disclosure: 

'We thank the Packard Foundation, the Curci Foundation, the Julia Burke Foundation, and other anonymous donors for their support of IGI FAST. We additionally thank the University of California, Berkeley for their financial support of IGI FAST.

A.J.E. is a graduate research fellow at the Greater Good Science Center at the University of California, Berkeley (https://greatergood.berkeley.edu/). J.R.H. is a Fellow

of The Jane Coffin Childs Memorial Fund for Medical Research

(https://www.jccfund.org/). A.M. is a fellow of the Damon Runyon Cancer Research Foundation (https://www.damonrunyon.org/).

The funders had no role in study design, data collection and analysis, decision to publish, or preparation of the manuscript.'

PLOS ONE requires you to include in your manuscript further information about the funder so that any relevant competing interests can be assessed. Please respond to the following questions:

a. Please state whether any of the research costs or authors' salaries were funded, in whole or in part, by a tobacco company (our policy on tobacco funding is at http://journals.plos.org/plosone/s/disclosure-of-funding-sources) 

b. Please state whether the donor has any competing interests in relation to this work (see http://journals.plos.org/plosone/s/competing-interests) .

c. Please state whether the identity of the donor might be considered relevant to editors or reviewers’ assessment of the validity of the work.

d. If the donors have no perceived or actual competing interests, please state: “The authors are not aware of any competing interests”.

This information should be included in your cover letter. We will amend your financial disclosure and competing interests on your behalf.

6. We note that you have a patent relating to material pertinent to this article.

a. Please provide an amended statement of Competing Interests to declare this patent (with details including name and number), along with any other relevant declarations relating to employment, consultancy, patents, products in development or modified products etc. Please confirm that this does not alter your adherence to all PLOS ONE policies on sharing data and materials, as detailed online in our guide for authors http://journals.plos.org/plosone/s/competing-interests by including the following statement: "This does not alter our adherence to  PLOS ONE policies on sharing data and materials.” If there are restrictions on sharing of data and/or materials, please state these.

Please note that we cannot proceed with consideration of your article until this information has been declared.

7. One of the noted authors is a consortium; IGI SARS-CoV-2 Testing consortium.

In addition to naming the author group, please list the individual authors and affiliations within this group in the acknowledgments section of your manuscript.

Please also indicate clearly a lead author for this group along with a contact email address.

8. Please include a separate caption for each figure in your manuscript.

9. Please include your tables 1-4 as part of your main manuscript and remove the individual files.

Please note that supplementary tables should remain as separate "supporting information" files.

10. Please include captions for your Supporting Information files at the end of your manuscript, and update any in-text citations to match accordingly. Please see our Supporting Information guidelines for more information: http://journals.plos.org/plosone/s/supporting-information

11. Your ethics statement should only appear in the Methods section of your manuscript. If your ethics statement is written in any section besides the Methods, please move it to the Methods section and delete it from any other section. Please ensure that your ethics statement is included in your manuscript, as the ethics statement entered into the online submission form will not be published alongside your manuscript.

Reviewers' comments:

Reviewer's Responses to Questions

**Comments to the Author**

1. Is the manuscript technically sound, and do the data support the conclusions?

Reviewer #1: Partly

Reviewer #2: Partly

2. Has the statistical analysis been performed appropriately and rigorously? 

Reviewer #1: I Don't Know

Reviewer #2: Yes

3. Have the authors made all data underlying the findings in their manuscript fully available?

Reviewer #1: Yes

Reviewer #2: Yes

4. Is the manuscript presented in an intelligible fashion and written in standard English?

Reviewer #1: Yes

Reviewer #2: Yes

5. Review Comments to the Author

Reviewer #1: 

This is a well-written manuscript describing an important, comprehensive approach to wide-scale testing, communication and social norming campaign on a university campus to test for and manage COVID-19. The authors developed a sophisticated, encrypted smartphone app for registering and scheduling sample collection, and models to aid in identifying populations to test. They also developed an important advertising and compliance campaign that greatly facilitates recruiting, compliance and participation in saliva screening on campus. They followed these activities with surveys that aided assessment of their testing program.

However, the authors cite four references from medRXiv, including their own, non-peer-reviewed manuscript detailing the testing protocol. This is completely inappropriate. The authors should either 1)include the testing protocol and data in this current manuscript for peer review or 2)submit the testing protocol manuscript for peer review then, upon acceptance, reference that study in this manuscript. They should also remove references to the other medRxiv manuscripts, as they are not peer reviewed and therefore cannot be used to support this manuscript currently under peer review. I am sorry I cannot be more supportive of this manuscript in its current form, and look forward to a revision that addresses these important concerns.

Reviewer #2: 

Throughout your paper, you state that a testing program such as yours can reduce or prevent outbreaks but you do not provide any evidence that your program did, in fact, reduce or prevent outbreaks. You either need to include evidence from your study that you did reduce/prevent outbreaks, or you should reframe your claims.

Consider rewording the first sentence of the introduction. The mere presence of viral genetic material does not necessarily correlate with infectivity and can lead to loss of productivity and impact to physical and mental health through unnecessary quarantine periods in non-infectious individuals.

Page 5 – what do you mean by “minimize worker use of PPE”?

Page 6 – how quickly were positives/inconclusives contacted by phone? What if they could not be reached?

Page 6 – why is it now deployed in a limited capacity? If it can, as you claim, prevent outbreaks, why has your own university not rolled it out on a larger scale.

What percentage of eligible on-campus staff decided to participate in the study?

What percentage of participants actually completed a test every second week from the time of their enrollment throughout the duration of the study (ie, full compliance)?

Please elaborate on your supposition that the exposure risk of your study cohort is lower than the general population of Berkley.

Only 5 positives were seen during your study. Were these positives asymptomatic or presymptomatic? Were their follow-up confirmatory tests positive or negative? At the time they were contacted by study personnel, had they already taken another COVID test? Did any outbreaks or clusters result from these individuals? How many infected individuals were traced back to these 5 positives, and was that number higher or lower than the Ro in the general Berkley community?

How many outbreaks occurred during this time period from people not undergoing voluntary testing?

You state that the responses to your survey indicate that a sense of civic duty and a cultural norm of frequent testing was present among your participants – is it not likely that those that chose to participate in this voluntary testing program were already individuals with a strong sense of civic duty/desire to undergo regular testing? How do you account for this selection bias in your conclusions?

Did you use risk stratification to guide your testing model? IE, you suggest continued brief surveys to determine risk profiles of the campus population, but did you ever trial or implement this in your study? If not, how do you propose another university apply the results of such continued brief surveys?

6. PLOS authors have the option to publish the peer review history of their article (what does this mean?). If published, this will include your full peer review and any attached files.

Reviewer #1: **Yes: **Mark Zabel

Reviewer #2: No

---

## [Author Response · Author response to Decision Letter 0]

19 Apr 2021

Dear Dr. Kalendar,

Thank you very much for the opportunity to resubmit our manuscript, “Launching a saliva-based SARS-CoV-2 surveillance testing program on a university campus” to PLOS ONE. We deeply appreciate the thoughtful responses and recommendations, which have together generated significant improvement to our manuscript. Below, we outline our responses to each point laid out in the reviews we received:

Editor’s comments

We have revised our manuscript’s styling to adhere to the referenced style requirements, including moving the methods section to immediately following the introduction.

2. Please ensure you have discussed any potential limitations of your study in the Discussion.

We now have a section entitled “Limitations” at the end of the “Results and Discussion” section.

3. In the Methods, please clarify that participants provided digital consent. Please also state in the Methods:

- Why written consent could not be obtained

- Whether the Institutional Review Board (IRB) approved use of digital consent

If additional written consent was collected as part of the test sample collection, please include this information.

For more information, please see our guidelines for human subjects research: https://journals.plos.org/plosone/s/submission-guidelines#loc-human-subjects-research

This information has been added to the methods to include:

“Recruitment, enrollment, consent, and participation for IGI FAST was approved by the Office for Protection of Human Subjects at the University of California, Berkeley under IRB #2020-05-13336. Informed consent and enrollment were completed on the IGI FAST web application instead of in writing, as a COVID-19 protocol to minimize the need for physical interaction. This web-based consenting step was approved by the IRB.”

No additional written consent was collected.

4. Thank you for including your ethics statement: 

"Office of the Protection of Human Subjects, University of California, Berkeley IRB #2020-05-13336. Written (online) consent was obtained from all participants". 

a. Please amend your current ethics statement to confirm that your named institutional review board or ethics committee specifically approved this study.

We have updated the Methods section as described in #3 above. This statement is also now listed in the “Ethics Statement” field of the submission form.

5. Thank you for stating in your financial disclosure: 

'We thank the Packard Foundation, the Curci Foundation, the Julia Burke Foundation, and other anonymous donors for their support of IGI FAST. We additionally thank the University of California, Berkeley for their financial support of IGI FAST.

A.J.E. is a graduate research fellow at the Greater Good Science Center at the University of California, Berkeley (https://greatergood.berkeley.edu/). J.R.H. is a Fellow of The Jane Coffin Childs Memorial Fund for Medical Research (https://www.jccfund.org/). A.M. is a fellow of the Damon Runyon Cancer Research Foundation (https://www.damonrunyon.org/).

The funders had no role in study design, data collection and analysis, decision to publish, or preparation of the manuscript.'

PLOS ONE requires you to include in your manuscript further information about the funder so that any relevant competing interests can be assessed. Please respond to the following questions:

a. Please state whether any of the research costs or authors' salaries were funded, in whole or in part, by a tobacco company (our policy on tobacco funding is at http://journals.plos.org/plosone/s/disclosure-of-funding-sources) 

b. Please state whether the donor has any competing interests in relation to this work (see http://journals.plos.org/plosone/s/competing-interests) .

c. Please state whether the identity of the donor might be considered relevant to editors or reviewers’ assessment of the validity of the work.

d. If the donors have no perceived or actual competing interests, please state: “The authors are not aware of any competing interests”.

This information should be included in your cover letter. We will amend your financial disclosure and competing interests on your behalf.

We amend our financial disclosure to the below. This has also been removed from the Acknowledgements section, per the request in #1:

We thank the Packard Foundation, the Curci Foundation, the Julia Burke Foundation, and other anonymous donors for their support of IGI FAST. We additionally thank the University of California, Berkeley for their financial support of IGI FAST. A.J.E. is a graduate research fellow at the Greater Good Science Center at the University of California, Berkeley (https://greatergood.berkeley.edu/). J.R.H. is a Fellow

of The Jane Coffin Childs Memorial Fund for Medical Research

(https://www.jccfund.org/). A.M. is a fellow of the Damon Runyon Cancer Research Foundation (https://www.damonrunyon.org/). The funders had no role in study design, data collection and analysis, decision to publish, or preparation of the manuscript. None of the research costs or authors’ salaries were funded in whole or in part by a tobacco company. The authors are not aware of any competing interests.

6. We note that you have a patent relating to material pertinent to this article.

a. Please provide an amended statement of Competing Interests to declare this patent (with details including name and number), along with any other relevant declarations relating to employment, consultancy, patents, products in development or modified products etc. Please confirm that this does not alter your adherence to all PLOS ONE policies on sharing data and materials, as detailed online in our guide for authors http://journals.plos.org/plosone/s/competing-interests by including the following statement: "This does not alter our adherence to PLOS ONE policies on sharing data and materials.” If there are restrictions on sharing of data and/or materials, please state these.

Please note that we cannot proceed with consideration of your article until this information has been declared.

The patents disclosed in the first submission of this article are unrelated to material pertinent to this article. As such, we are removing any mention of such patents from our submission.

7. One of the noted authors is a consortium; IGI SARS-CoV-2 Testing consortium.

In addition to naming the author group, please list the individual authors and affiliations within this group in the acknowledgments section of your manuscript.

Please also indicate clearly a lead author for this group along with a contact email address.

This has been updated

8. Please include a separate caption for each figure in your manuscript.

This has been provided, as outlined in the documents referenced in #1.

9. Please include your tables 1-4 as part of your main manuscript and remove the individual files.

Please note that supplementary tables should remain as separate "supporting information" files.

This has been provided

10. Please include captions for your Supporting Information files at the end of your manuscript, and update any in-text citations to match accordingly. Please see our Supporting Information guidelines for more information: http://journals.plos.org/plosone/s/supporting-information

These have been added.

11. Your ethics statement should only appear in the Methods section of your manuscript. If your ethics statement is written in any section besides the Methods, please move it to the Methods section and delete it from any other section. Please ensure that your ethics statement is included in your manuscript, as the ethics statement entered into the online submission form will not be published alongside your manuscript.

This has been updated accordingly.

Reviewer #1

12. …the authors cite four references from medRXiv, including their own, non-peer-reviewed manuscript detailing the testing protocol. This is completely inappropriate. The authors should either 1)include the testing protocol and data in this current manuscript for peer review or 2)submit the testing protocol manuscript for peer review then, upon acceptance, reference that study in this manuscript. They should also remove references to the other medRxiv manuscripts, as they are not peer reviewed and therefore cannot be used to support this manuscript currently under peer review.

We thank this reviewer for their thoughtful feedback on our manuscript.

In citing the preprint manuscripts, we are following guidelines provided by PLOS “...preprints are a citable part of the scientific record. All preprints are given a permanent DOI, which should be used when adding to the reference list of a manuscript.” This is consistent with other prominent journals. These guidelines indicate that the journal does consider citation of medRxiv articles to be appropriate. With this, we aim to be transparent in our application of the methodologies or conclusions provided by the pre-print.

Of the four medRxiv citations, at least two are presently under consideration at peer-reviewed journals, including one at PLOS ONE. We hope that they will be accepted prior to acceptance of this manuscript, allowing for citation of peer-reviewed versions. Providing further details regarding these manuscripts would be beyond the scope of the present manuscript.

For the citation (Silva et al) where we provide reference to the manuscript’s conclusions, we have added “...in pre-print literature…” to the sentence where we make the citation.

The other citations describe the application of methods or technology as described by the other authors.

Reviewer #2

13. Throughout your paper, you state that a testing program such as yours can reduce or prevent outbreaks but you do not provide any evidence that your program did, in fact, reduce or prevent outbreaks. You either need to include evidence from your study that you did reduce/prevent outbreaks, or you should reframe your claims.

We are grateful to this reviewer for their thoughtful feedback and recommendations for areas of improvement.

Insomuch as this paper lacks a proper comparison of outbreaks in a similar population without regular surveillance testing, we agree with this reviewer’s assessment. As such, we have reframed our claims as follows:

“To test this model, optimize our assay…” is now “To test the operational feasibility of this model, optimize our assay…”

“Here we describe a blueprint for a low-barrier, safe, effective, easy, and adaptable program for campus SARS-CoV-2 surveillance using saliva specimens capable of preventing outbreaks and effectively creating a culture of safety.” now reads ”Here we describe a blueprint for a low-barrier, safe, effective, easy, and adaptable program for campus SARS-CoV-2 surveillance using saliva specimens capable of minimizing the number of outbreaks [9] and effectively creating a culture of safety.” Additionally, it now cites [9], justifying the statement that our paradigm is capable of minimizing the number of outbreaks.

While now reframed, we maintain our reference to outbreak prevention given that we demonstrated our capacity to detect asymptomatic or presymptomatic individuals and asymptomatic or presymptomatic individuals infected with SARS-CoV-2 are capable of transmitting SARS-CoV-2 infection to others

14. Consider rewording the first sentence of the introduction. The mere presence of viral genetic material does not necessarily correlate with infectivity and can lead to loss of productivity and impact to physical and mental health through unnecessary quarantine periods in non-infectious individuals.

We appreciate the point the reviewer is raising. Accordingly, we have amended the first sentence of the introduction to clarify: “...when the pandemic features asymptomatic or presymptomatic infectious individuals.”

15. Page 5 – what do you mean by “minimize worker use of PPE”?

This is meant to illuminate a benefit of self-administered sample collection - the lack of demand of PPE needed for safe collection. Traditional nasopharyngeal swabs require items such as face shields and gowns, which are not required for self-collected saliva.

We have reworded this to “...minimize worker demand of…”

16. Page 6 – how quickly were positives/inconclusives contacted by phone? What if they could not be reached?

We have amended “Participants with positive or inconclusive results were additionally contacted via phone by one of the study clinicians” to include “...within minutes to several hours following the lab reporting the result to the clinicians.” As with normal clinical resulting for COVID-19, the clinicians continued to attempt to contact the participants by phone until the call was received. Additionally, these participants received the result via encrypted email.

17. Page 6 – why is it now deployed in a limited capacity? If it can, as you claim, prevent outbreaks, why has your own university not rolled it out on a larger scale.

As stated in the following paragraph under “Testing and participant characteristics”, “Overall, IGI FAST featured a high number (n=761, 6.4%) of “specimen insufficient” results, making it a difficult test to further scale through pooled testing [2].”

To clarify this point in the location this reviewer refers to, “Instead of saliva, a self-administered nasal swab tested on the same PCR-based platform as the IGI FAST test is used for widespread regular asymptomatic surveillance testing at the University of California, Berkeley because these swab-based samples were more easily pooled than the saliva samples.”

18. What percentage of eligible on-campus staff decided to participate in the study?

Unfortunately, the University’s estimate of the total on-campus population for the duration of this study is highly imprecise. As such, we are unable to estimate the percentage of on-campus staff that were actively participating.

19. What percentage of participants actually completed a test every second week from the time of their enrollment throughout the duration of the study (ie, full compliance)?

We have added:

“648 (17.7%) of the 3,653 active participants participated fully (i.e., took every appointment opportunity offered to them) (Fig 6). The reasons underlying the lack of full participation by 82.3% of participants were not systematically assessed; however, anecdotal accounts by participants cited periods of not conducting on campus work, travel, or self-direction towards clinical testing due to symptoms or exposure as some reasons for missing appointments.”

20. Please elaborate on your supposition that the exposure risk of your study cohort is lower than the general population of Berkley.

We have elaborated the referenced sentence:

“...however, we speculate that it likely reflects a difference in exposure risk between our study cohort and the broader population of the City of Berkeley.“

to

“...however, we speculate that it likely reflects a difference in demographics and associated exposure risk between our campus’ study cohort and the broader population of the City of Berkeley.“

21. Only 5 positives were seen during your study. Were these positives asymptomatic or presymptomatic? Were their follow-up confirmatory tests positive or negative? At the time they were contacted by study personnel, had they already taken another COVID test? Did any outbreaks or clusters result from these individuals? How many infected individuals were traced back to these 5 positives, and was that number higher or lower than the Ro in the general Berkley community?

How many outbreaks occurred during this time period from people not undergoing voluntary testing?

Our IRB did not provide for follow up with participants to gather clinical or contact tracing information. Instead, the purpose of IGI FAST was to establish a program by which asymptomatic or presymptomatic individuals could be routed to clinical services. As such, we are unable to address any of these questions.

We have brought attention to this limitation in a new section of the Results and Discussion section “Limitations”.

22. You state that the responses to your survey indicate that a sense of civic duty and a cultural norm of frequent testing was present among your participants – is it not likely that those that chose to participate in this voluntary testing program were already individuals with a strong sense of civic duty/desire to undergo regular testing? How do you account for this selection bias in your conclusions?

We have included mention of this in the Limitations section. Additionally, we have changed:

“We conclude that with appropriate messaging and atmosphere, surveillance testing for SARS-CoV-2 can quickly become a social norm and even be perceived as a civic duty.”

to

“Acknowledging that our survey lacks assessment of individuals not choosing to participate in surveillance testing, we conclude that with appropriate messaging and atmosphere, surveillance testing for SARS-CoV-2 can quickly become a social norm and even be perceived as a civic duty.”

23. Did you use risk stratification to guide your testing model? IE, you suggest continued brief surveys to determine risk profiles of the campus population, but did you ever trial or implement this in your study? If not, how do you propose another university apply the results of such continued brief surveys?

Towards the beginning of the section discussing risk-stratification, we clarify:

“Such an allocation model was not deployed in IGI FAST, as the program was initially designed solely for a population of on-campus workers.”

We have also amended:

“Doing so would layer adaptive risk-stratification onto a baseline risk-stratification paradigm to ensure adequate resource allocation longitudinally.”

To

“Doing so, with the assistance of data-driven epidemiological models such as Brook et al [9], would layer adaptive risk-stratification onto a baseline risk-stratification paradigm to ensure optimal resource allocation longitudinally.”

Sincerely,

Jennifer Doudna, PhD

---

## [Decision Letter · Decision Letter 1]

26 Apr 2021

Launching a saliva-based SARS-CoV-2 surveillance testing program on a university campus

PONE-D-21-05507R1

Dear Dr. Doudna,

We’re pleased to inform you that your manuscript has been judged scientifically suitable for publication and will be formally accepted for publication once it meets all outstanding technical requirements.

Kind regards,

Ruslan Kalendar, PhD

Academic Editor

PLOS ONE

Reviewers' comments:

Reviewer's Responses to Questions

**Comments to the Author**

1. If the authors have adequately addressed your comments raised in a previous round of review and you feel that this manuscript is now acceptable for publication, you may indicate that here to bypass the “Comments to the Author” section, enter your conflict of interest statement in the “Confidential to Editor” section, and submit your "Accept" recommendation.

Reviewer #1: All comments have been addressed

Reviewer #2: All comments have been addressed

2. Is the manuscript technically sound, and do the data support the conclusions?

Reviewer #1: Yes

Reviewer #2: Yes

3. Has the statistical analysis been performed appropriately and rigorously? 

Reviewer #1: Yes

Reviewer #2: Yes

4. Have the authors made all data underlying the findings in their manuscript fully available?

Reviewer #1: Yes

Reviewer #2: Yes

5. Is the manuscript presented in an intelligible fashion and written in standard English?

Reviewer #1: Yes

Reviewer #2: Yes

6. Review Comments to the Author

Reviewer #1: (No Response)

Reviewer #2: All of my comments have now been addressed, thank you for your responses and revisions to this manuscript.

7. PLOS authors have the option to publish the peer review history of their article (what does this mean?). If published, this will include your full peer review and any attached files.

Reviewer #1: **Yes: **Mark D Zabel

Reviewer #2: No

---

## [Editor Report · Acceptance letter]

17 May 2021

PONE-D-21-05507R1 

Launching a saliva-based SARS-CoV-2 surveillance testing program on a university campus 

Dear Dr. Doudna:

I'm pleased to inform you that your manuscript has been deemed suitable for publication in PLOS ONE. Congratulations! Your manuscript is now with our production department. 

Kind regards, 

on behalf of

Prof. Ruslan Kalendar 

Academic Editor

PLOS ONE